# Pedestrian Safety in Frontal Tram Collision, Part 2: Laminated Glass as a Crucial Part of the Absorption and Deformation Zone—Its Impact Test and Analysis

**DOI:** 10.3390/s23218974

**Published:** 2023-11-04

**Authors:** Roman Jezdik, Marek Sebik, Petr Kubovy, Frantisek Marsik, Frantisek Lopot, Barbora Hajkova, Dita Hylmarova, Martin Havlicek, Ondrej Stocek, Martin Doubek, Tommi Tikkanen, Martin Svoboda, Karel Jelen

**Affiliations:** 1VUKV a.s., 158 00 Prague, Czech Republic; jezdik@vukv.cz; 2SVS FEM s.r.o., 628 00 Brno, Czech Republic; msebik@svsfem.cz; 3Department of Anatomy and Biomechanics, Charles University, 162 52 Prague, Czech Republic; 4Institute of Thermomechanics, Czech Academy of Sciences (CAS), 182 00 Prague, Czech Republic; 5Department of Designing and Machine Elements, Czech Technical University, 160 00 Prague, Czech Republic; martin.havlicek@fs.cvut.cz (M.H.);; 6Dopravní Podnik Hlavního Města Prahy, 190 22 Prague, Czech Republic; doubekm@dpp.cz; 7GIM Oy, 02650 Espoo, Finland; 8Faculty of Mechanical Engineering, Jan Evangelista Purkyne University, 400 96 Usti nad Labem, Czech Republic; martin.svoboda@ujep.cz; 9Department of Pathophysiology, Second Faculty of Medicine (2. LF UK), Charles University, 150 06 Prague, Czech Republic; 10Police Academy of the Czech Republic in Prague, 143 00 Prague, Czech Republic

**Keywords:** tram, crash test, front face, windshield, pedestrian, safety, model, head injury criterion, laminated glass, impact test, Ansys optiSLang, Ansys LS-DYNA

## Abstract

As was shown in the previous part of the study, windshields are an important part of the passive safety means of modern low-floor trams with an extraordinary effect on pedestrian safety in a pedestrian–tram collisions. Therefore, maximum attention must be paid to the definition of tram windshield characteristics. This article describes a windshield crash test, from which data are obtained to verify the feasibility of the applied computational approaches. A developed analytical model is utilised for a simple description of the energy balance during collision with an illustrative definition of the important parameters of laminated glass as well as their clear physical interpretations. The finite element analysis (FEA) performed in Ansys software using two versions of material definition, namely a simpler (*MAT_ELASTIC with nonlocal failure criterion) and a more complex (*MAT_GLASS with brittle stress-state-dependent failure) material model, which are presented as suitable for obtaining a detailed description of the shattering process of laminated glass, which can also be used effectively in windshield engineering.

## 1. Introduction

This article is based on the findings of the previous article (Part 1), where the importance of the windshield as crucial part of the deformation zone of the front face of a tram in a collision with a pedestrian was identified and demonstrated in actual crash tests (Part 1).

The issue of tram front collisions with pedestrians is currently being discussed by the European Technical Committee CEN/TC 256. The discussion has already resulted in a technical report (see [1]). Usually, an adult pedestrian’s head and/or shoulders come into contact with the windshield in the event of a serious frontal collision with a modern low floor urban tram. Consequent head/brain injuries can be monitored using electrical impedance tomography [2]. Figure 1 shows examples of actual fatal accidents that took place on 4 November 2019 in Prague and on 2 February 2019 in Dresden.

Laminated glass is widely used as windshield material in the automotive and railway urban vehicle industry. It is a simple sandwich composite structure often called “safety glass” because of its excellent impact energy absorption and the mitigation of glass fragment dispersion. The windshield normally consists of two glass sheets bonded together by means of one interlayer usually made of polyvinyl butyral (PVB) foil.

The study presented below further elaborates on the aforementioned topic. The aim of this article is to provide explanations and possible descriptions of existing laminated glass properties during a crash, which will be applicable to the further development and engineering of the material.

In order to clearly explain the principle of the impactor penetration process through the glass, an analytical model has been developed that has simple physical quantities—stiffness and viscosity. This approach makes all individual components of the model easy to understand and also gives them a clear physical interpretation. And in the end, it enables the very simple quantification of the most important parameters of the described glass by means of quantities that are comprehensible to both the engineering and production community.

For a detailed analysis of the process of glass breaking during a collision, it is necessary to use numerical methods. Finite element modelling (FEM) is the most commonly utilised method. The article presents two computational models based on the use of definitions of the varied complexity of the used material models and damage criteria accounting for the real composition of the investigated glass layers. The purpose of this analysis was to find a reliable model and to identify the key parameters of the material models under consideration and damage criteria that would be suitable for the engineering description of laminated glass behaviour.

All presented computational approaches are based on and verified through experimental data. An important output of this article is the description of a suitable procedure for experimental data acquisition, which can provide results that are directly applicable for the characterisation of computational material models.

## 2. Experimental Data Acquisition—The Impact Test

Although there is a plethora of published works on road vehicle collisions, most of them deal with occupants, i.e., persons inside the vehicle, or the collisions of pedestrians with cars but not trams (e.g., [5,6,7]). In general, it might be difficult to use conventional dummies to study collision situations because this necessitates the wide positioning of extremities against the position of the main body and also makes specific demands on their construction to avoid the distortion of data during the experiment [8]. For a number of situations, mainly in automotive industry, dummies can thus not be directly and easily used [9]. To obtain reliable and transferrable experimental data, the creation of stable and repeatable measurement conditions is a must. Thus, many methodological overview publications are available on this topic (e.g., [10,11]. After the careful consideration of the above aspects, we decided to use a well-defined impactor as a representation of a pedestrian’s head in our experiments.

The setup of the impact test is given in Figure 2. An impactor with a spherical impact surface was hinged on a pair of steel ropes. The impactor was made of steel. Its mass was 5.5 kg. The radius of the impact surface was 75 mm. The impactor was placed at a predetermined height and fixed using a thin string. Cutting the string started movement of the impactor against the laminated glass. The movement of the impactor takes place in the vertical plane parallel with the track. The contact point of the impactor and the glass was about 300 mm above the lower edge of the window and about 165 cm above the rail. This point approximately corresponds to expected contact point of the tram glass with head peace of the Hybrid-III 50th Percentile Male standing anthropomorphic test device. An accelerometer was installed on the impactor to measure impactor acceleration. The sampling frequency of the accelerometer was 100 kHz. A high-speed camera provided a side view to record the trajectory and velocity in the plane of the impactor movement. It functioned using a sampling frequency of 20 KHz, which was important for the detailed observation of the glass and foil behaviour during the impact. The impact speed in the tests was 5.64 m/s. 

The windshield tested was made of two glass sheets. Each of the sheets had a thickness of 3.00 mm. The sheets were bonded together using polyvinyl butyral foil with a thickness of 0.76 mm.

## 3. Simplified Model of Glass Double-Layer Deflection Plate

The viscoelastic plate consists of double glass, which is connected via a transparent elastic foil and embedded in a rigid frame as, shown in Figure 3. 

The balance of inertial and external surface forces at each material point of the viscoelastic plate of the volume $V_{g}$ with density $\rho =2550\;\mathrm{kg}m^{-3}$ and with the surface of the glass plate $A_{g}$ in the case of an incident body is expressed via the following equation:(1)$$\int_{V_{g}}\rho {\dot{v}}^{i}dv=\int_{V_{g}}\frac{\partial t^{ki}}{\partial x^{k}}dv=\int_{A_{g}}t^{ki}n_{k}da,\;\mathrm{for}\;i,k=1,2,3$$

The balance of forces for each material point of the plate can be written as
(2)$$\rho {\frac{dv}{dt}}^{i}=\frac{\partial t^{ik}}{\partial x^{k}},{\;v}^{i}={\frac{du}{dt}}^{i}...\mathrm{deformation}\;\mathrm{rate}\;u^{i}=x^{i}-X^{i}...\mathrm{displacement}\;\mathrm{of}\;\mathrm{the}\;\mathrm{material}\;\mathrm{point}\;\;\mathrm{relative}\;\mathrm{to}\;\mathrm{the}\;\mathrm{initial}\;\mathrm{state}\;X^{i}.$$

This equation expresses the balance of the inertial force (gravitational force is neglected), and the surface force is expressed by the stress tensor.
(3)$$t^{ij}=t_{el}^{ij}+t_{dis}^{ij}=\underset{t_{el}^{ij}}{\underbrace{Ke_{\left(1\right)}\delta^{ij}+2\bar{\mu }\overset{\left(o\right)}{e^{ij}}}}+\mu_{v}d_{\left(v\right)}\delta^{ij}+\underset{t_{dis}^{ij}}{\underbrace{2\mu \overset{\left(o\right)}{d^{ij}}}}\;\mathrm{for}\;e_{\left(1\right)}=\frac{\partial u^{i}}{\partial x^{i}}=\mathrm{div}\;u,\overset{\left(o\right)}{e^{ij}}=\frac{1}{2}\left(\frac{\partial u^{i}}{\partial x^{j}}+\frac{\partial u^{j}}{\partial x^{i}}-\frac{2}{3}e_{\left(1\right)}\delta^{ij}\right)$$

We separated pure volume changes $e_{\left(1\right)}$ and shape changes $e_{ij}^{\left(o\right)}=e_{ij}-e_{\left(1\right)}\delta_{ij}/3$, which is the so-called distortion. The relationship of Young’s modulus of elasticity E and Poisson’s number σ to Lamé coefficients λ,K is [13]
(4)$$\lambda =\frac{E\sigma }{\left(1+\sigma \right)\left(1-2\sigma \right)},K=\frac{E}{3\left(1-2\sigma \right)},\bar{\mu }=\frac{E}{2\left(1+\sigma \right)},c_{t}=\sqrt{\frac{\bar{\mu }}{\rho }}=\sqrt{\frac{E}{2\rho \left(1+\sigma \right)}}.$$

Young’s modulus of glass and Poisson’s number at room temperature are E=50−90 GPa and σ=0.23, respectively, while $c_{t}$ is the velocity of propagation of weak transverse waves.

For glass filled with polyester matrix, the following values are used: E=17.2 GPa, Poisson number $\sigma_{pr\;}=0.45$, and density $\rho_{pr}=1.2\times 10^{3}\;\mathrm{kg}m^{3}$.

The strain rate tensor
(5)$$d_{ij}={\dot{e}}_{ij},\overset{\left(o\right)}{d_{ij}}=d_{ij}-\frac{1}{3}{\dot{e}}_{\left(1\right)}\delta_{ij}=\frac{1}{2}\left(\frac{\partial v_{i}}{\partial x^{j}}+\frac{\partial v_{j}}{\partial x^{i}}-\frac{2d_{\left(1\right)}}{3}\delta_{ij}\right),d_{\left(1\right)}=\mathrm{div}v$$
is needed to describe the viscous properties of the material. Viscosity describes damping in general, and we distinguish between shear viscosity μ and bulk viscosity $\mu_{\left(v\right)}$. The latter viscosity is neglected in our case. With a uniaxial load in the direction of the axis, the shear rate is equal.
(6)$$\frac{d(z_{0}-z)}{\Delta rdt}=\frac{\partial v_{z}}{\partial r}\approx \frac{\partial^{2}\varsigma }{\partial r\partial t}$$

The formation of cracks and the subsequent permanent deformation of the material is modelled by viscosity, so that the viscous part of the stress is equal
(7)$$t_{dis}^{zr}=\mu \frac{d\Delta z}{\Delta rdt}∼\mu \frac{\partial^{2}\varsigma }{\partial r\partial t}$$
where $\mu \left[\mathrm{Pa}s\right]$ is the Newtonian viscosity. The viscosity of the uncracked glass is of the magnitude $\mu =1.2\times 10^{10}\;\mathrm{Pa}s$. However, in the case of cracks, although its exact value is unknown, it may have a major effect on the permanent deformation of the glass plate. Rubbers at high deformation rates $10^{5}\;s^{-1}$ have a viscosity in the order of 100 Pa s.

In the force balance Equation (2), we discard the convective term $v_{j}\frac{\partial u^{i}}{\partial x^{j}}$ and consider purely elastic disturbances, which are described only by the elastic part of the stress tensor $t_{el}^{ij}$, see Equations (3) and (4).

We can divide the problem into two parts:(i)the steady part of the impact accompanied by a large deformation with subsequent damage to the glass plate (formation of cracks)(ii)the dynamic response of an elastic plate to a sudden impact characterized by the plate vibration and with the propagation of transverse waves in the plate. The result of this part is the time course of the force (acceleration) acting on the impacting body.

### 3.1. The Total Deformation of the Plate

Given the low speed of the impacting body, we can omit the inertia of the glass. (The speed of vibration is much greater than the speed of impact.) We use the relationship of the deflection of a circular plate embedded in the centre of the glass and the stationary concentrated force $F_{c}$.

Using the theory of linear elasticity, based on the elastic strain tensor (3), the amount of deflection ς(r) of the embedded circular plate at the point of force can be written as [14,15]:(8)$$F_{c}=\frac{8\pi h^{3}E}{3\left(1-\sigma \right)\left(3+\sigma \right)R^{2}}{\left.\zeta \right|}_{\frac{r}{R}=0}=\frac{K_{h}h^{3}}{R^{2}}{\left.\zeta \right|}_{\frac{r}{R}=0},\;\mathrm{for}\;K_{h}=\frac{8\pi E}{3\left(1-\sigma \right)\left(3+\sigma \right)}$$

We assume that these are two glass plates with a radius *R* = 0.3 m, each with a thickness *h* = 3.45 mm and material parameters of σ=0.23, $E=70\;GPa,\;\mathrm{and}K_{h}=235\;Gpa$. In terms of mechanical resistance, the glass plates can slide over each other.

The total kinetic energy of the incident body is transformed into a deformation of magnitude $\zeta_{cel}$. Assuming that the deformation energy is equal to the kinetic energy of the mass body, $m_{B}=5.5\;\mathrm{kg}$, moving with the velocity, $v_{B}=5.64\;ms^{-1}$, we obtain
(9)$$\begin{array}{l} E_{B,kin}=\frac{m_{B}v_{B}^{2}}{2}=\int_{0}^{\varsigma_{cel}}F_{c}\left(\xi \right)d\xi +2\pi \mu \int_{0}^{\zeta_{cel}}\int_{0}^{R}\frac{\partial v_{z}}{\partial r}rdrd\zeta =\frac{K_{h}\zeta_{cel}^{2}}{2}+1.96\mu v_{B}R\zeta_{cel}\\ =A\xi_{cel}^{2}+B\xi_{cel},\;\mathrm{where}\;A=\frac{K_{h}h^{3}}{2R^{2}},B=1.96\mu v_{B}R \end{array}$$

The force $F_{c}$ acts on the trajectory $\zeta_{cel}$ and causes a corresponding part of the elastic deformation (energy). The dissipative (damped) part of the energy is included in the cracked glass only. The formation of cracks in the element containing the viscosity $\mu \left[\mathrm{Pa}s\right]$ is included, analogously to Equation (7), as the force $F_{S}$ in Figure 3. The viscosity introduced in this way is only theoretical, and its aim is to account for the irreversible deformation of the glass. The deformation (cracking) of the glass has a decisive influence, which under certain conditions absorbs all elastic energy. The deceleration course of the incident body is affected by the total displacement at the point of impact (deformation path), which can be determined from Equation (9). The general relationship for deflection after the impact is
(10)$$\zeta_{cel}={\left.\sqrt{\frac{E_{kin}}{A}}\left[\sqrt{1+\frac{B^{2}}{4AE_{kin}}}\right]-\frac{B}{2A}\right|}_{B^{2}<<4AE_{kin}}=\sqrt{\frac{E_{kin}}{A}}+\frac{B}{2A}\left(\frac{B}{4\sqrt{AE_{kin}}}-1\right)$$

In the case of omitting the cracking of the glass, i.e., without viscosity, the elasticity is dominant (variable A). All energy is absorbed into the elastic deformation, the magnitude of which is
(11)$$\zeta_{el}=\sqrt{\frac{E_{kin}}{A}}=\sqrt{\frac{m_{B}}{h^{3}K_{h}}}Rv_{B}=40\mathrm{mm}$$

In the physical experiment, a value of $\zeta_{cel,act}=63\mathrm{mm}$ was measured. This difference reflects the size of the damping variable B and provides the possibility to estimate the limit of the irreversible damage to the glass. Under the conditions
(12)$$B=1.96\mu v_{B}R\geq 4\sqrt{AE_{kin}},\mathrm{or}\mu ≐\frac{K_{h}\left(\xi_{cel,act}-\xi_{el}\right)}{v_{B}}{\left(\frac{h}{R}\right)}^{3}=1.4\times 10^{3}\mathrm{Pa}s$$
cracks occur, and the total deformation according to Equation (10) is equal to the deformation $\zeta_{cel,act}=63\mathrm{mm}$, which was experimentally determined.

### 3.2. Generation of Transverse Waves on the Plate

After the impact of the cylindrical plate, transverse vibrations are generated that oscillate in the *z* direction and propagate from the centre to the radius r and symmetrically in the direction ϕ (see Figure 3). Regarding the momentum balance (Equation (2)), and while considering Equation (8), we will assume that the deformation of the plate in the z direction will also depend on time, i.e., $u_{z}\left(r,\phi ,t\right)=\varsigma (r,t)$.

The balance of forces (Equation (2)) during the impact of a mass body $m_{B}$ on a deformable surface at the point of contact is obtained through the following equation:(13)$$m_{B}\ddot{\varsigma }+F_{s}+F_{c}=0,\;\mathrm{where}\;{\dot{v}}_{z}=\ddot{\varsigma }\left(r,t\right)\;\mathrm{is}\;\mathrm{the}\;\mathrm{acceleration}.$$

We express the effect of shear stresses (Equation (7)) by force, which is obtained through:(14)$$F_{s}=4\pi hR\mu {\left.\frac{\partial^{2}\varsigma }{\partial r\partial t}\right|}_{r=R}=\hat{\mu }{\dot{\varsigma }}_{r},\;\mathrm{for}\;{\dot{\varsigma }}_{r}=\frac{\partial \dot{\varsigma }}{\partial r}$$
where ${\dot{\varsigma }}_{r}$ is the effect of the stress rate gradient (see Figure 3). The effect of damping (viscosity, formation of cracks) is expressed using the empirical parameter:(15)$$\hat{\mu }=4\pi hR\mu \;\left[\mathrm{Pa}sm^{2}=Ns\right]$$
which is related to the dynamic viscosity μ in the material relationship, when considering the relevant sample geometry. After substituting for the external force (Equation (8)) and internal force (Equation (14)), we obtain a linear differential equation:(16)$$m_{B}\ddot{\varsigma }+\hat{\mu }{\dot{\varsigma }}_{r}+K_{g}\varsigma =0,\;\mathrm{for}\;{\;K}_{g}=\frac{h^{3}K_{h}}{R^{2}}$$

The contact force $F_{c}$ is considered here as a response to the elastic deformation of an external force. The balance of forces expressed by Equation (13) represents a damped oscillator with initial conditions.
(17)$$\varsigma (t=0)=0,\dot{\;\varsigma }(t=0)=v_{B}$$

The velocity of the impacting body is $v_{B}$. We are looking for a solution in the form of a travelling wave along the plate in the direction r, i.e., from the point of impact of the body towards the edge of the board.
(18)$$\varsigma \left(r,t\right)=\varsigma_{0}\exp\left[i\left(\omega t-k_{r}r\right)\right],{\;k}_{r}=k_{r,re}+ik_{r,im}$$

Here, $k_{r}$ is a r component of a wave vector and ω is the frequency. The perturbation amplitude $\varsigma_{0}$ will be determined using the steady solution (Equation (11)) and the energy balance (Equation (9)). By substituting into the force balance of Equation (16), we obtain the dispersion equation:(19)$$\omega^{2}m_{B}-\hat{\mu }\omega \left(k_{r,re}+ik_{r,im}\right)-K_{g}=0,\;\mathrm{for}\;\omega =\omega_{re}+i\omega_{im}$$
by solving this, we find the connection between the frequency of the vibration $\omega_{re}=2\pi /t$, its wavenumber $k_{r,re}=2\pi /\lambda_{r}$, and the respective damping, which includes the term $\omega_{im}$. Their mutual relationship is described by the following equation:(20)$$\omega =\frac{\hat{\mu }\left(k_{r,re}+ik_{r,im}\right)}{2m_{B}}\pm \frac{1}{2m_{B}}\sqrt{{\hat{\mu }}^{2}\left(k_{r,re}^{2}-k_{r,in}^{2}\right)+4m_{B}K_{g}+i{\hat{\mu }}^{2}2k_{r,re}k_{r,im}}$$

The general form of the solution is
(21)$$\zeta \left(r,t\right)=C_{1}e^{-\omega_{im}t}e^{-k_{r,im}r}\sin\left(\omega_{re}t-k_{r,re}r\right)+C_{2}e^{-\omega_{im}t}e^{-k_{r,im}r}\cos\left(\omega_{re}t-k_{r,re}r\right)$$

These are damped waves on the surface of the plate that move at speed $\omega_{re}/k_{r,re}$.

#### 3.2.1. Generation of Pure Transverse Elastic Waves on the Plate

We limit ourselves to the purely elastic deformation obtained by omitting the damping term, i.e., $\hat{\mu }=0$. The analysis of the effect of viscosity (the formation of cracks) on the time course of plate vibrations is outside the scope of this article. The plate vibrates as an undamped harmonic oscillator with Eigen frequencies:(22)$$\omega_{re}=\frac{2\pi }{T}=\sqrt{\frac{K_{g}}{m_{B}}}=\sqrt{\frac{K_{h}h^{3}}{m_{B}R^{2}}}=\sqrt{\frac{235\times 10^{9}\times {\left(3.45\times 10^{-3}\right)}^{3}}{5.5\times 0.3^{2}}}=140{\;s}^{-1}$$

The excitation of the transverse vibrations of the plate tightly clamped at the edges is clearly visible. The whole plate vibrates at this frequency, the vibration period is $T=2\pi /\omega_{re}=45\mathrm{ms}$. This frequency describes the dynamic parameters of the board and allows us to determine elastic deflection and acceleration.
(23)$$\begin{array}{l} \varsigma \left(t\right)=\varsigma_{0}\sin\omega_{re}t,\dot{\;\varsigma }\left(t\right)=\omega_{re}\varsigma_{0}\cos\omega_{re}t,\ddot{\varsigma }(t)=-\omega_{re}^{2}\varsigma_{0}\sin\omega_{re}t\\ \mathrm{for}\;\mathrm{initial}\;\mathrm{condition}\;\varsigma \left(t=0\right)=0,\dot{\varsigma }(t=0)=v_{B}=\omega_{re}\varsigma_{0} \end{array}$$

We thus obtain an additional relationship between the maximum deflection and the impact velocity. Compared with Equation (11), we see that in the case of purely elastic changes, the amplitude of the deflection is the same as the maximum deflection:(24)$$\varsigma_{0}=\frac{v_{B}}{\omega_{re}}=v_{B}\sqrt{\frac{m_{B}}{K_{g}}}=39\;\mathrm{mm}≐\varsigma_{el}$$
and the deformation of the whole plate is
(25)$$\varsigma (t)=\varsigma_{el}\sin\omega_{re}t=39\sin\left(140t\right)$$

Thus, the maximum deviation is reached for the value $\omega_{re}t_{max}=\frac{\pi }{2}$, which corresponds to time $t_{max}=11\;ms$. The time obtained by the experiment is approximately 2 times larger, i.e., 25 ms. The reason for the difference will probably be material failure (the formation of cracks) and subsequent permanent deformation, following the general solution (Equation (21)). Given the initial condition, the constant $C_{2}=0.$ The travelling wave (Equation (21)) has a maximum at $\omega_{re}t-k_{r,re}r=\frac{\pi }{2}.$ For $k_{r,re}=\frac{2\pi }{\lambda_{r}}$ and $r=\lambda_{r}/4$, we have $t_{max}=\pi /\omega_{re}=22\;ms$. This conclusion confirms the course of the acceleration (it has the same course as the force)
(26)$$\ddot{\varsigma }(t)=-\omega_{re}^{2}\varsigma_{0}\sin\omega_{re}t=-78g\sin\omega_{re}t$$

Figure 4 shows the acceleration (in the case of an impact, it is the deceleration of the impacting body) even for the initial phase of the impact, when the plates are not yet fully disconnected. The stiffness of the plates is higher and the corresponding natural frequency of oscillation of the plate (Equation (22)) is higher too.

#### 3.2.2. Propagation of Transverse Waves Inside a Plate

After the impact from the cylindrical plate, transverse waves (deformation in the z direction) are generated as axially symmetric and propagate in the r  direction from the centre (see Figure 3 for ϕ=φ). Considering Equations (2) and (3), we obtain the wave equation for the transverse waves of the small amplitude $u_{z}\left(r,\phi ,t\right)$ in the z direction equation:(27)$$\frac{\partial^{2}u_{z}}{\partial^{2}t}=c_{t}^{2}\left(\frac{\partial^{2}u_{z}}{\partial r^{2}}+\frac{\partial u_{z}}{r\partial r}+\frac{\partial^{2}u_{z}}{r^{2}\partial \phi^{2}}\right),\;\mathrm{for}\;c_{t}^{2}=\frac{E}{2\rho \left(1+\sigma \right)}$$
where $c_{t}$ is the transverse wave velocity (see Equation (4)). We assume a time-periodic solution of the form of
(28)$$u_{z}=U_{z}\left(r\right)\Phi \left(\phi \right)e^{i\omega t}$$
and after substituting it into the wave (Equation (27)), we obtain
(29)$$-\omega^{2}U_{z}\Phi =c_{t}^{2}\left(\frac{\Phi d^{2}U_{z}}{dr^{2}}+\frac{\Phi dU_{z}}{rdr}+\frac{U_{z}d^{2}\Phi }{r^{2}d\phi^{2}}\right)$$

We assume that the function is periodic (perturbations circulate around the centre at radius r) so it is a solution to the following equation:(30)$$\frac{d^{2}\Phi }{d\phi^{2}}+n^{2}\Phi =0,\;e.g.,\;\Phi \left(\phi \right)=\Phi_{0}e^{ik_{\phi }\phi }\mathrm{for}k_{\phi }=n$$

By substituting into Equation (29), and after modification, we obtain
(31)$$\left\{\frac{d}{rdr}\left(\frac{rdU_{z}}{dr}\right)+\left[{\left(\frac{\omega }{c_{t}}\right)}^{2}-\frac{n^{2}}{r^{2}}\right]U_{z}\right\}\Phi =0$$
and, for any non-zero function Φ, we obtain the Bessel equation
(32)$$\frac{d}{rdr}\left(\frac{rdU_{z}}{dr}\right)+\left[{\left(\frac{\omega }{c_{t}}\right)}^{2}-\frac{n^{2}}{r^{2}}\right]U_{z}=0$$

By changing the variable, we can write this equation in the usual dimensionless form:(33)$$\frac{d}{\tilde{r}d\tilde{r}}\left(\frac{\tilde{r}dU_{z}}{d\tilde{r}}\right)+\left[1-\frac{n^{2}}{{\tilde{r}}^{2}}\right]U_{z}=0,\;\mathrm{for}\;{\;U}_{z}\left(\tilde{r}\right),\mathrm{where}\tilde{r}=\frac{\omega }{c_{t}}r$$

This equation describes the propagation of transverse waves in a plate with radius R. At the edge of the fixed plate, there is zero deflection $U_{z}\left(R\right)=0$, and the solution of Equation (27) is given by the superposition of the solution of n modes and has the form
(34)$$U_{z}\left(\tilde{r}=\frac{\omega }{c_{t}}r\right)=\sum_{n=1}A_{n}J_{n}\left(\tilde{r}\right)$$
where n is the order of the Bessel function. For the basic mode n=0 (the waves propagate symmetrically from the point of impact of the body), only the function $J_{0}\left(\tilde{r}\right)$ is sufficient. For this mode, the solution (Equation (28)) of the wave Equation (27) is given using a simplified relationship:(35)$$u_{z}(\frac{\omega_{0}}{c_{t}}r,t)=u_{z0}J_{0}\left(\frac{\omega_{0}}{c_{t}}r\right)\sin\omega_{0}t$$
where $u_{z0}$ is the initial deflection amplitude. The longest wavelength $\lambda_{r}$ of this oscillation is given by the distance of the edge from the impact point and is equal to $\lambda_{r}=4R$ for R=0.3 m (see Figure 3). The relationship to the wave frequency $\omega_{0}=2\pi /\tau_{0}$ is determined by the zero point of the Bessel function $J_{0}\left(\tilde{r}\right)=0,\mathrm{for}\tilde{r}\approx 2.4$, so that
(36)$$\tilde{r}\approx 2.4=\frac{\omega_{0}}{c_{t}}R=\frac{\omega_{0}}{c_{t}}\frac{\lambda_{r}}{4},\;\mathrm{and}\;{\;\omega }_{0}=2.4\frac{c_{t}}{R}$$
and for transverse wave velocity(Equation (4))
(37)$$c_{t}=\sqrt{\frac{E}{2\rho \left(1+\sigma \right)}}=\sqrt{\frac{55\times 10^{9}}{2\times 2550\left(1+0.25\right)}}=2937\;ms^{-1}$$
we thus obtain
(38)$$\tau_{0}=\frac{2\pi R}{2.4c_{t}}=0.27\times 10^{-3}\;s$$

The time of the first deflection is equal to $\tau_{0}/4=0.68\times 10^{-4}s$. At approximately this time, the maximum contact force occurs. The cause of this peak is probably the formation of the first larger crack in the glass plate (see Figure 4) and is therefore decisive for determining the HIC criterion (compare with Equation (26)).

Remark: For a rectangular plate it would be $\tilde{r}=\pi /2$, since the fundamental mode would be described by the function $\cos\tilde{r}$ and the zero point of this function is $\tilde{r}=\pi /2$.

## 4. Numerical Simulations

### 4.1. Modelling Approach

The finite element analysis (FEA) is one of the most advanced approaches for modelling a laminated glass impact test today [16]. This approach incorporates various constitutive laws and failure models, captures elastic and plastic shock wave propagation, and does not require any model symmetry. These qualities of the FEA lead to high suitability and the easy applicability of FEA for this kind of analysis.

Common simulation models use the brittle material model for glass. The material model of PVB interlayer is not entirely unambiguous. The elasto-plastic, hyper-elastic, viscoelastic, or visco-plastic material models can be taken into consideration. The key problem of the impacted laminated glass behaviour modelling is the modelling of its damage. The principal damage pattern, glass–ply cracking, can be modelled using several numerical algorithms, which are described in detail in [16], namely the element deletion method (EDM), the continuum damage mechanics (CDM), the discrete element method (DEM), the combined discrete/finite element methods (DEM/FEMs), the extended finite element method (XFEM), and the cohesive zone model (CZM).

Methods utilising one, two, or three layers of elements are used to model laminated glass behaviour. The first method uses layered composite properties to represent the entirety of the laminated glass. The second method uses one layer of shell elements for the glass and one layer of membrane elements for the PVB-layer with shared nodes. Timmel et al. [17] introduced a smeared modelling technique that utilised a simple two-layer composite model. The three-layer model is more physically realistic because it corresponds to the three-layer sandwiched structure of laminated glass. Shell or solid elements are used for the modelling of glass and membrane, and solid or shell elements can be used for the discretisation of the PVB-layer. For really detailed modelling, attention should also be paid to the issue of delamination, which reveals a differently complex character involving local delamination, which can cause considerable non-stationarity in the course of the monitored quantities [18].

Since the glass impact test includes dynamic loading, the nonlinear response of the structure, large deformation, and the overall duration is only about 40 ms, and the explicit FEA was chosen. The software used in this study for FEA was Ansys LS-DYNA ver. 2020/R2 (Ansys, Canonsburg, PA, USA), as it is able to capture all the listed event phenomena [19].

A new material model for laminated glass in the LS-Dyna software, *MAT_280 (*MAT_GLASS) [20], which replaces the previously used material *MAT_032 (*MAT_LAMINATED_GLASS), is introduced. Paper [21] describes dynamic mechanical behaviour of laminated windshields subjected to head-form impact and compares two models; single-layered model with *MAT_032 and triple-layered model with *MAT_123 (*MAT_MODIFIED_PIECEWISE_LINEAR_PLASTICITY). The material model *MAT_280 uses isotropic linear elastic material law with brittle failure for glass and supports crack closure and opening effects. Failure is handled without removing elements. It is easy to use, the computation time is shorter, and it can be used in crash simulations. The disadvantage is that it uses a local failure criterion that is not suitable for the first acceleration peak modelling (see [20]).

### 4.2. Finite Element Analysis (FEA)

FE simulations in this study are also focused on the same impact event on a laminated glass plate, as in Figure 5. While investigating the appropriateness of various modelling approaches, two different FE modelling techniques were used and compared. The main difference was the choice of the glass material model. Both methods considered the laminated structure as a three-layered circular plane. All three layers are uniformly meshed using four-node Belytschko–Tsay shell elements with five thickness integration points and an edge length of 6 mm. Uniform mesh without symmetry boundary conditions and without any special mesh adjustments (such as arranging mesh edges in radial and tangential directions around the centre of impact) is a prerequisite for the versatile utilisation of the model as the centre of impact is generally unknown beforehand. The mesh size was designed to also allow for the utilisation of this model in other crash/impact FE models without any significant timestep drops. All degrees of freedom of the nodes at the borders of the glass plate were fixed. Bonding between individual layers was prescribed using *CONTACT_TIED_SHELL_EDGE_TO_SURFACE_BEAM_OFFSET_ID, which ensures that neither the separation nor the relative movement of the layers can occur. As for the loading, the impactor was modelled as a rigid object discretised with solid hexahedral elements. The impactor was given corresponding inertia properties and an initial velocity of 5.64 m/s and was placed in front of the glass plate.

The material models chosen for the glass are:

Model 1:Linear elastic material model with nonlocal failure criterion.Model 2:“Glass” material model with brittle, stress-state-dependent failure criterion (*MAT_GLASS).

The material model of the PVB foil is simple bilinear plastic in both cases. In the next steps, the suitability and variability of these two approaches were further subjected to sensitivity analysis and optimisation in Ansys optiSLang.

### 4.3. Model 1

In this model, the material behaviour of the glass was defined using the linear elastic material model with the following material parameters: elastic modulus *E =* 70 GPa, density *ρ =* 2.5 × 10^−6^ kg/mm^3^, and Poisson’s ratio σ = 0.23. The linear elastic material model was extended with the nonlocal failure criterion with the LS-DYNA keyword *MAT_ADD_EROSION. This failure criterion was in this context designed as an original and is mainly intended for windshield impact. The criterion is defined using the parameters of critical energy, critical radius, and critical first principal stress. When the critical first principal stress is exceeded at any element, the failure mechanism is triggered. The element is marked as the centre of impact. Then, an energy criterion is monitored at all elements inside a circle, defined through a critical radius and the centre of impact. The energy criterion is met when the internal energy of a shell element exceeds the adjusted critical energy (critical energy multiplied by the area factor). Then, the element is deleted. Usage of this criterion is limited to shell elements [22].

#### 4.3.1. Sensitivity Analysis

At first, the sensitivity analysis of this model was performed. For this analysis, the sensitivity to failure criterion parameters and foil stiffness parameters were studied. The impact simulation was monitored using parameters that reflect the agreement with the experimental results. Signal data from the experiment were simplified into piecewise linear functions to capture the main character and eliminate signal noise.

The input parameters were varied within the prescribed ranges according to the AMOP (adaptive metamodel of optimal prognosis) design of experiments scheme [22]. In total, the sensitivity analysis was evaluated based on 300 design points. When material failure did not occur, designs exhibited much higher (and unrealistic) stiffness and were excluded from the analysis. The results indicated that the most influential input parameters (the highest correlation parameter (CoP) values [22]) are the glass critical energy and foil tangent modulus. The sensitivity analysis served also as a pre-optimisation. 

Typically, designs with good displacement agreement did not match force signal as well and vice versa (see Figure 6a,b). The best design that is a trade-off between all observed aspects must be found.

#### 4.3.2. Optimisation Analysis

Due to the preceding sensitivity analysis, the relationships between individual objectives can be understood and pre-optimised design can be used as a starting point. The optimisation was aimed at three aspects of the simulation: *u_max_dif*, *F_max_dif*, and *F_flat_max_dif*. The evolutionary algorithm was utilised, and 200 design points were evaluated in total. Similarly to the results of the sensitivity analysis, it was possible to identify designs that performed well with respect to a single objective but was considerably worse with respect to the others. In the end, design #84 (see Table 1) was found to be the optimal trade-off on the Pareto front.

### 4.4. Model 2

In this model, the material behaviour of the glass was defined with *MAT_GLASS. The material behaviour before failure is an isotropic, small strain linear elasticity with Young’s modulus, *E*, and Poisson’s ratio, σ. Asymmetric (tension–compression-dependent) failure occurs as soon as one of the following plane stress failure criteria is violated: Rankine maximum stress, Mohr–Coulomb, or Drucker–Prager. As soon as failure occurs in the tensile regime, a crack appears perpendicularly to the maximum principal stress direction (the crack coordinate system is set up and stored). The subsequent process of reduction in stress and stiffness tensor components is governed by a chosen failure criterion (Rankine, Mohr–Coulomb, or Drucker–Prager) [22]. The model can consider up to two orthogonal cracks per integration point, simultaneous failure over element thickness, and crack closure effects. Therefore, no additional failure criteria are needed. This advanced material model requires up to 22 input parameters. 

#### 4.4.1. Sensitivity Analysis

As with Model 1, the sensitivity analysis was performed first. The outputs of the simulations were monitored in the same manner as with Model 1. The design of experiments method was advanced Latin hypercube and there were 500 design points evaluated in total. Again, designs with no failure were subsequently excluded from the analysis. Because of the presence of nominal discrete parameters in this analysis, the processing of more complex relations was not possible. In terms of linear correlation, the parameters of tensile strength and glass critical energy influence the simulation results the most. 

The most promising designs found during the analysis were chosen as starting points for subsequent optimisation.

#### 4.4.2. Optimisation Analysis

To make the optimisation more effective, the multi-objective optimisation was transformed into single objective optimisation. The objective was to minimise a function which considers all monitored aspects with a weight function based on sensitivity analysis:(39)$$f=\sum \frac{\left|X_{i}^{exp}-X_{i}^{sim}\right|}{X_{i}^{exp}}\cdot w_{i},\;\mathrm{for}\;\mathrm{scalar}\;\mathrm{parameters}$$
(40)$$f=\sum \frac{{\left\|X_{i}^{exp}-X_{i}^{sim}\right\|}_{2}}{{\left\|X_{i}^{exp}\right\|}_{2}}\cdot w_{i},\;\mathrm{for}\;\mathrm{signals}$$
where *Xiexp/sim* represents the *i-th* monitored scalar parameter or signal from a simulation or experiment and *w_i_* is a weight function (Table 2).

The evolutionary algorithm found several very good designs quickly. In total, 100 designs were evaluated. One of the most common attributes of the best designs was the Mohr–Coulomb failure criterion. The best design (#67, see Table 3) matched both displacement and force measurement.

## 5. Conclusions

Considering the previous part of the study (Part 1), probably the most important message of this article is the confirmation of the significant influence of PVB foil placed between the glass layers to slow down the motion of a pedestrian and reduce the impact peak to their head. The influence of the PVB layer can be observed on the distance necessary for the deceleration of the impactor, which can be seen in Figure 7 and Figure 8. Of course, certain portions of the impact energy are also absorbed by the shattering of individual glass layers. In the actual versions of the windshield glass, this positive effect is largely outweighed by high rigidity and elasticity modulus values, which cause high impact loading in the initial phase of the collision. This is why the PVB foil is in current general shape in tram front faces is the most important structural element in terms of pedestrian safety. 

Despite the substantial simplification of the situation and despite the fact that the presented analytical approach uses a very simple computational model based on the combination of stiffness and viscosity of individual layers of the windshield, it is possible in this way to easily determine the necessary properties of the individual layers of the windshield through engineering applicable quantities. By introducing viscosity, the analytical solution offers a simplified model of the permanent deformation of cracked glass. By fitting the magnitude of this viscosity, the theoretical model gives the value of glass deflection $\zeta_{cel,act}=63\;mm$ (see Equation (10)). The corresponding value given via simulation (Model 2, design #67) is $u_{max}=65\;mm$ (see Figure 8a). According to both the theoretical model of glass cracking (see Equation (21) and (25)) and simulation (see Figure 8a), the maximum deflection is reached at time $t_{max}$ = 22 ms. According to the experiment, this time is equal to 25 ms, due to real shape of the measured windshield.

Transverse wave propagation from the point of impact is modelled using the Bessel function $J_{0}$, which describes concentric waves. The propagation of these waves has been observed experimentally. The occurrence of the first glass crack can be determined from their frequency and speed, see Equations (36) and (37). It appears approximately after two oscillations of the plate, i.e., at time 0.6 ms after the impact (compare Equation (38) and Figure 4).

Similarly, the FE approach also proved effective in Section 4. in which the best matching material model parameters were found and compared to experimental results.

An important step in the FEA in terms of the reliability and validity of its results is to perform sensitivity analysis and optimisation (for Ansys in OptiSLang). In the direct comparison of the FEA results with experimental data, the model using *MAT_GLASS material definition achieves better correspondence than the model that uses the simple *MAT_ELASTIC material definition with nonlocal failure criterion. This difference is probably caused by the fact that *MAT_GLASS definition also considers crack closure.

From the above findings, it can be concluded that the material parameters obtained from a simple crash test on the one hand and those found through optimisation analyses the using described simplified model on the other hand can be used as initial parameters for the detailed optimisation both of the computational model and the physical specimen of the full windshield with respect to its geometry and the overall design of the front face of the tram.

## Figures and Tables

**Figure 1 sensors-23-08974-f001:**
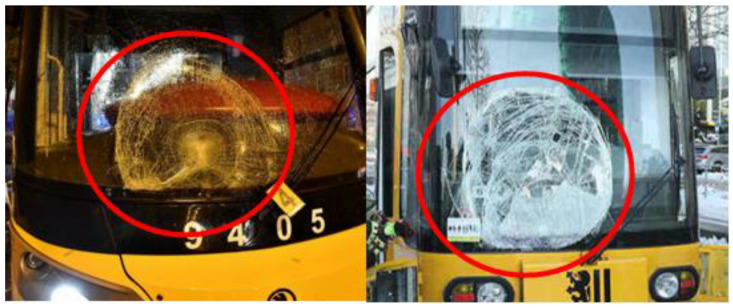
Windshield after collision with a pedestrian: Prague, 4 November 2019 (**left**) [3]; Dresden, 2 February 2019 (**right**) [4].

**Figure 2 sensors-23-08974-f002:**
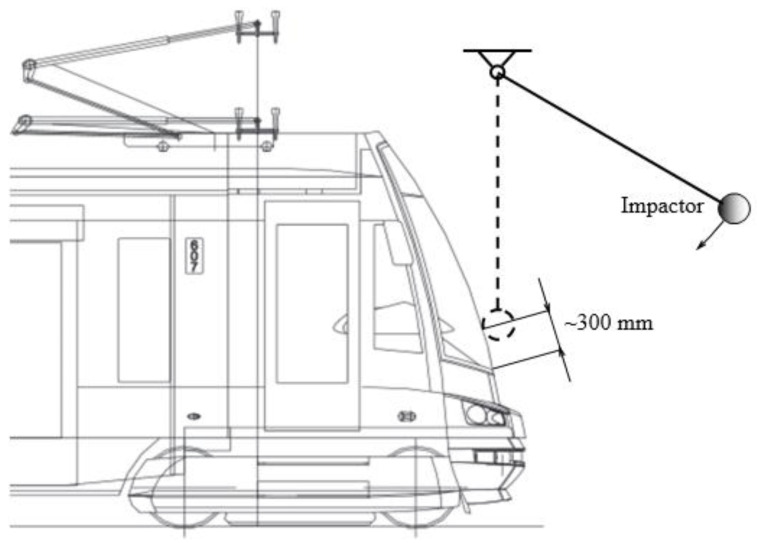
Impact test setup [12].

**Figure 3 sensors-23-08974-f003:**
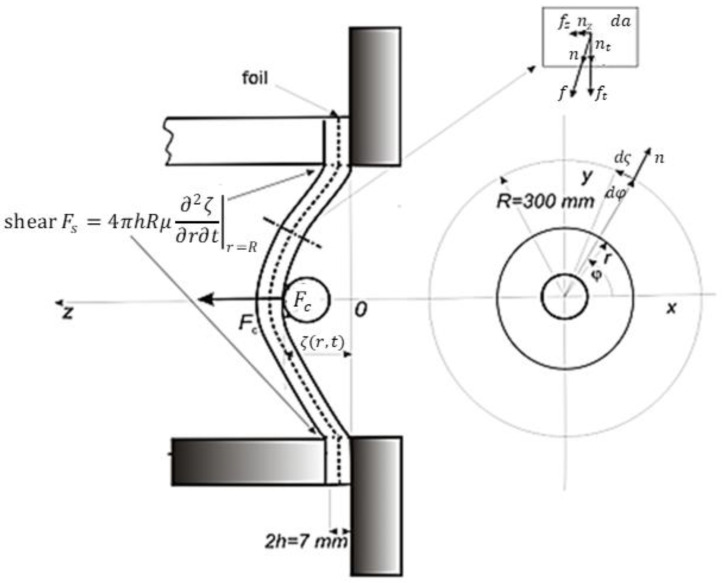
Model setup for the test described above.

**Figure 4 sensors-23-08974-f004:**
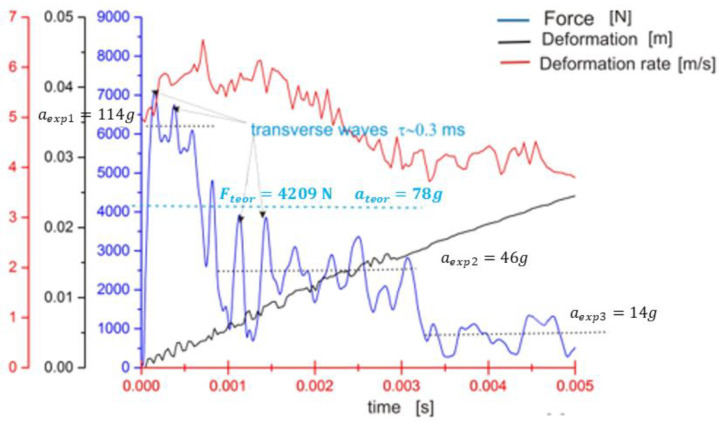
Detailed recording of the board’s vibrations and transverse wave propagation at the moment of impact. The shot was 0.3 m from the edge. For a fixed edge of glass, the longest wavelength of the transverse wave in the plate oscillation is 1.2 m. The time between two transverse waves is approx. 0.27 ms, see (Equation (38)).

**Figure 5 sensors-23-08974-f005:**
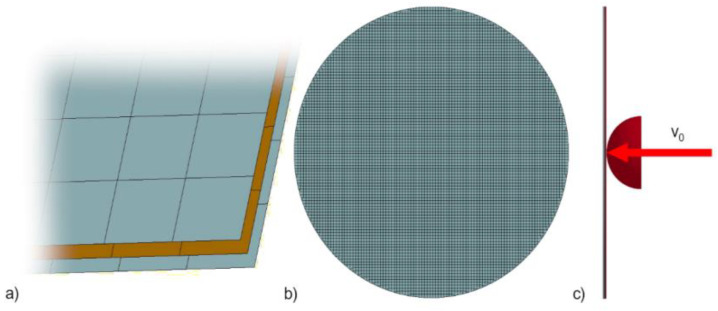
FE simulation setup: (**a**) detail of the three-layer composite as the model of the laminated glass; (**b**) complete model of the studied glass–circular plane; (**c**) side view of the impact model of the hemispherical impactor on the studied glass.

**Figure 6 sensors-23-08974-f006:**
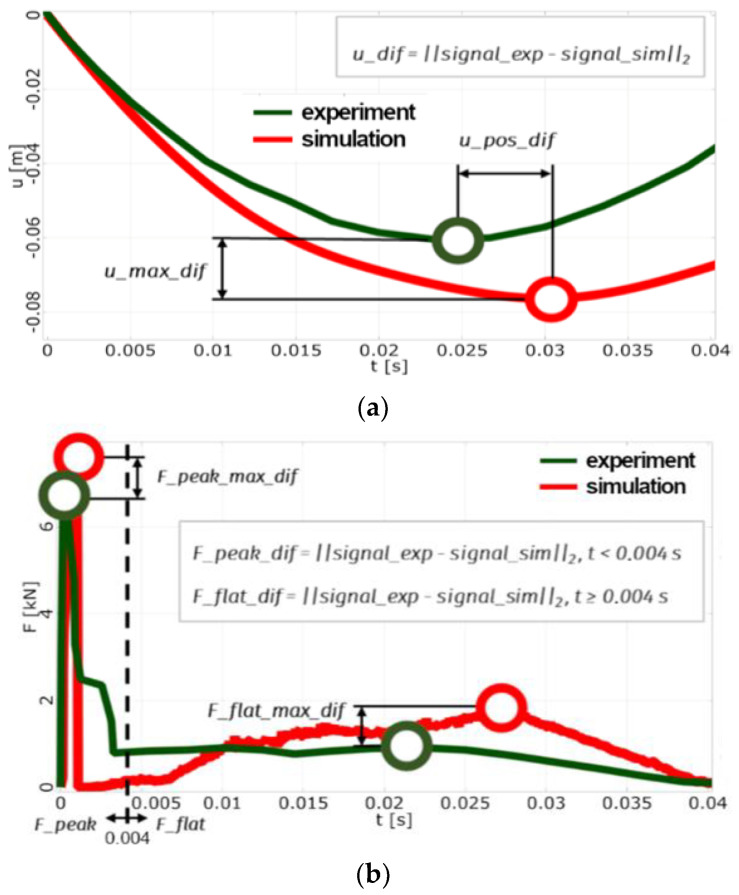
(**a**) Simulation–experiment comparison method: displacement vs. time *(u_max_dif* is the difference between the maximum displacements obtained by experiment and simulation and *u_pos_dif* is the difference between position (in time) of maximum displacement obtained through experiment and simulation). (**b**) Simulation–experiment comparison method: force vs. time (*u_max_dif* is the difference between the maximum displacements obtained through experiment and simulation and *u_pos_dif* is the difference between the position (in time) of maximum displacement obtained through experiment and simulation).

**Figure 7 sensors-23-08974-f007:**
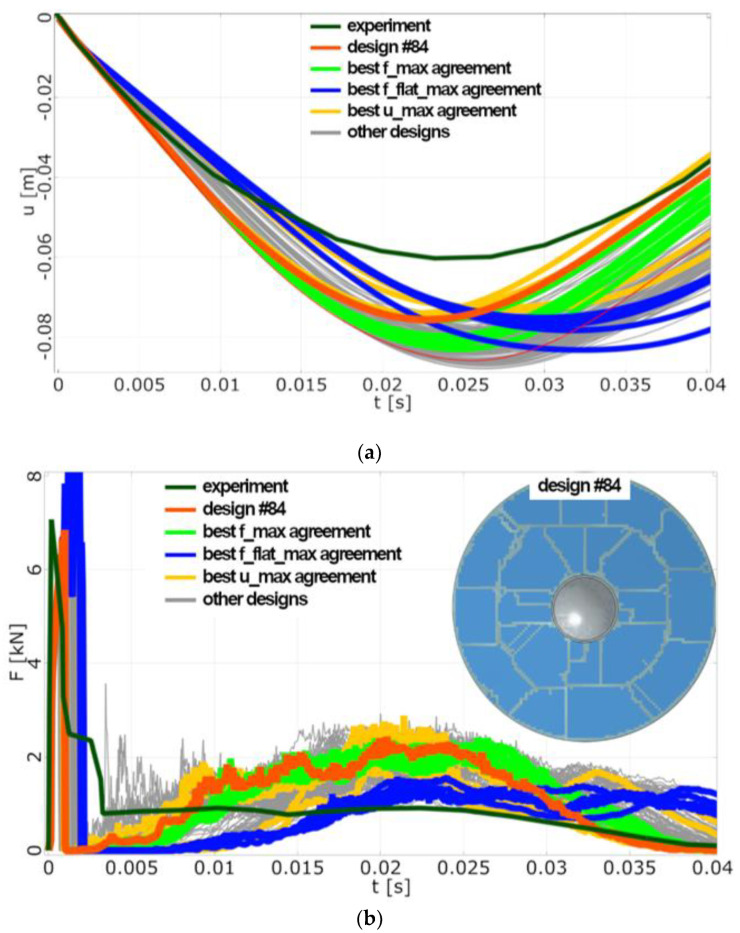
(**a**) FE Model 1, optimisation results: displacement vs. time. (**b**) FE Model 1, optimisation results: force vs. time.

**Figure 8 sensors-23-08974-f008:**
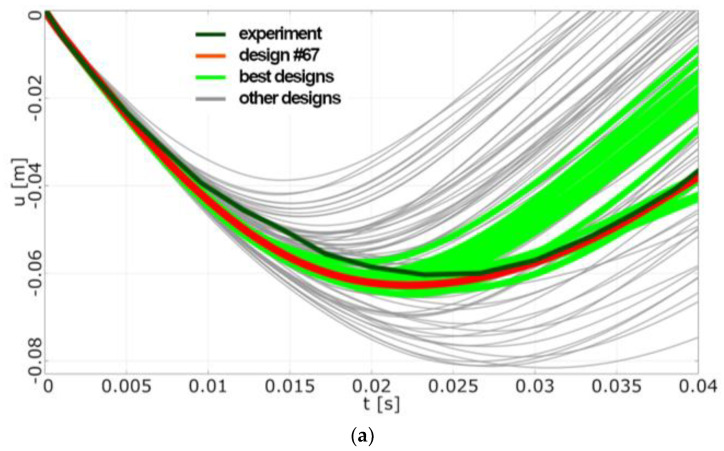
(**a**) FE Model 2, optimisation results: displacement vs. time. (**b**) FE Model 2, optimisation results: force vs. time.

**Table 1 sensors-23-08974-t001:** Material model parameters of the best design (#84).

Quantity	Symbol	Unit	Value
Density	*RO (ρ)*	kg/m^3^	2500
Young’s modulus	*E*	GPa	70
Poisson’s ratio	PR (σ)	-	0.23
Maximum principal stress at failure	*SIGP1*	GPa	0.0793
Critical energy for nonlocal failure criterion	*ENGCRT*	J	397
Critical radius for nonlocal failure criterion	*RADCRT*	mm	1900

**Table 2 sensors-23-08974-t002:** Weight function for monitored aspects.

*X_i_*	*F_flat*	*F_flat_max*	*F_max*	*F_peak*	*u*	*u_pos*	*u_max*
*w_i_*	0.1	0.5	7	0.5	1	1	10

**Table 3 sensors-23-08974-t003:** Material model parameters of the best design (#67).

Quantity	Symbol	Unit	Value
Density	*RO (ρ)*	kg/m^3^	2500
Young’s modulus	*E*	GPa	70
Poisson’s ratio	*PR* (σ)	-	0.23
Flag for failure criteria	*IFMOD*	-	1
Tensile strength	*FT*		0.0609
Compression strength	*FC*	GPa	1
Damage parameter	*AT*	-	0.838
Damage parameter	*BT*	-	0.00888
Damage parameter	*AC*	-	1
Damage parameter	*BC*	-	1000
Scale factor for the tensile strength	*FTSCL*	-	0.704
Scale factor for stiffness after failure	*SFSTI*	-	0.000583
Scale factor for stress in case of failure	*SFSTR*	-	0.0774
Flag for crack strain initialization	*CRIN*	-	0
Crack strain to reactivate stress components	*ECRCL*	-	0.00115
Number of cycles for stress reduction	*NCYCR*	-	2
Number of failed int. points	*NIPF*	-	1
Critical value for element deletion	*EPSCR*	-	−0.1
Critical energy for nonlocal failure criterion	*ENGCRT*	J	0.0247
Critical radius for nonlocal failure criterion	*RADCRT*	mm	1370
Quasi-static strain rate threshold	*RATENL*	mm/(mm∙ms)	10^−6^
Smoothing factor on the effective strain rate	*RFILTF*	-	0.9

## Data Availability

The data presented in this study are available on request from the corresponding author with the permission of partners of the project. The data are not publicly available due to their technical meaning for partners of the project.

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
