# Peer review of "Pedestrian Safety in Frontal Tram Collision, Part 2: Laminated Glass as a Crucial Part of the Absorption and Deformation Zone—Its Impact Test and Analysis"

_sensors, 2023, doi:10.3390/s23218974_

Round 1
Reviewer 1 Report
PVB foil in glass layers is studied for impact peaks. The scientific purpose of this part is clear. Some minor questions are as below:
Where do the material behaviors of glass come from?
More explanation of failure model in FEM by Ls-dyna should be given, refer to 10.1007/s12206-022-0609-5
The difference of displacement between experiment and simulation (Figure 6) generally should not be so large. Please check why.
The force and displacement are used as the design criteria. Is it better to consider the damage?
Some of the sections have too many paragraphs.
Author Response
Dear reviewer,
many thanks to you for all your helpful comments and recommendations. Please, find enclosed file with my reactions and revised article resubmitted to the website.
Kind regards
Frantisek Lopot

Reviewer 2 Report
The reviewer did not find any discussion/comparison of the simple model described in Chapter 3 and the FE model from Chapter 4. The analysis made in Chapter 4 is not novel ( by itself), the authors also should disseminate the differences between the standard windshield testing in the automotive industry and their approach to tram windshield testing. Each material model described in Chapter 4 should be described in table form in the text or in the attachment. It should be also clearly described if the simulation was made in Ansys LS-DYNA, or in the LSTC LS-DYNA ( it is especially important because the LSTC was bought by Ansys company ( please also add the version of the solver).
Author Response

(The authors gave the same response as above.)

Round 2
Reviewer 2 Report
Thank you for the improvement. The paper is acceptable in its current form.
Author Response
Dear reviewer,
thank you again for your very helpful feedback on our work - we have learned a lot.
Kind regards
Frantisek Lopot